# Synthetic Micro/Nanomotors for Drug Delivery

**Eduardo Guzmán** [1,2,*] and **Armando Maestro** [3,4]

1   Departamento de Química Física, Facultad de Ciencias Químicas, Universidad Complutense de Madrid, Ciudad Universitaria s/n, 28040 Madrid, Spain
2   Instituto Pluridisciplinar, Universidad Complutense de Madrid, Paseo Juan XXIII 1, 28040 Madrid, Spain
3   Centro de Física de Materiales (CSIC, UPV/EHU), Paseo Manuel de Lardizabal 5, 20018 San Sebastián, Spain
4   IKERBASQUE—Basque Foundation for Science, Plaza Euskadi 5, 48009 Bilbao, Spain
*   Correspondence: eduardogs@quim.ucm.es; Tel.: +34-9-1394-4107

**Abstract:** Synthetic micro/nanomotors (MNMs) are human-made machines characterized by their capacity for undergoing self-propelled motion as a result of the consumption of chemical energy obtained from specific chemical or biochemical reactions, or as a response to an external actuation driven by a physical stimulus. This has fostered the exploitation of MNMs for facing different biomedical challenges, including drug delivery. In fact, MNMs are superior systems for an efficient delivery of drugs, offering several advantages in relation to conventional carriers. For instance, the self-propulsion ability of micro/nanomotors makes possible an easier transport of drugs to specific targets in comparison to the conventional distribution by passive carriers circulating within the blood, which enhances the drug bioavailability in tissues. Despite the promising avenues opened by the use of synthetic micro/nanomotors in drug delivery applications, the development of systems for in vivo uses requires further studies to ensure a suitable biocompatibility and biodegradability of the fabricated engines. This is essential for guaranteeing the safety of synthetic MNMs and patient convenience. This review provides an updated perspective to the potential applications of synthetic micro/nanomotors in drug delivery. Moreover, the most fundamental aspects related to the performance of synthetic MNMs and their biosafety are also discussed.

**Keywords:** biomedicine; drug delivery; machines; micro/nanomotors; nanomedicine; propulsion





## 1. Introduction

Molecular biological motors have inspired research on the seeking of self-propelled supramolecular engines, so-called synthetic micro/nanomotors (MNMs). These offer a broad range of applications in different fields, including drug delivery, environmental remediation, biosensing, and precision surgery at the micro-/nanoscale. In fact, synthetic micro/nanomotors are structures characterized by reduced dimensionality and have the ability to convert energy obtained from diverse sources into kinetic energy, i.e., into motion. Therefore, it is of a paramount importance to deepen the understanding of the mechanisms driving the energy conversion in synthetic MNMs to ensure an accurate control over the motion of the manufactured devices [1]. This is essential for the fabrication of suitable micro/nanomotors for biomedical applications.

The potential biomedical applications of synthetic micro/nanomotors have opened new avenues for overcoming some of the current challenges of precision medicine, e.g., low permeation of the biological barriers, towards a more efficient and personalized treatment of different diseases [2]. Furthermore, synthetic micro/nanomotors add to the characteristics of traditional drug delivery systems, e.g., drug protection, selectivity and biocompatibility, and the capability of swimming and penetration through cellular barriers [1]. Therefore, it may be expected that synthetic MNMs present the capability for encapsulation, transport, and supply of drugs directly to the disease sites, which in turn contributes to enhancing

the therapeutic efficiency and decreasing the systemic side effects associated with the administration of toxic drugs [3].

There are available a broad range of fabrication methods enabling the fabrication of synthetic micro/nanomotors with different chemical composition and geometries, e.g., nanowires, microtubular microrockets, Janus microspheres, and supramolecular-based motors [4]. Nevertheless, the most common micro/nanomotors are miniaturized human-made engines characterized by an asymmetric structure and/or chemical heterogeneity. This enables the rupture of the thermodynamic balance, which must be considered an essential aspect for driving the conversion of the energy from chemical reactions or external sources to mechanical motion [5–7]. This motion results from an intricate balance between the drag force and the propulsion, operating over the micro/nanomotors, which guides the motion displacement under low Reynolds number conditions [8,9]. This is possible by using different propulsion mechanisms, based on both endogenous (i.e., chemotaxis) or exogenous (e.g., ultrasound, magnetic fields, light) stimuli, which can allow simultaneously the transport of drugs to specific targets and a triggered release at the right time [10,11]. Moreover, MNMs can be also functionalized using different strategies, enabling their use for in vivo imaging [12]. Therefore, nanotechnology and nanomaterials can contribute to finding solutions to several challenges in the precise therapy for different diseases, and the design of smart synthetic micro/nanomotors has opened up exciting opportunities that can contribute to solving complex problems that are not always easy to address with conventional approaches [13].

The last decade has been very fruitful in the fabrication of synthetic micro/nanomotors with high biocompatibility, multifunctionality, and efficient propulsion in biological fluids, which has provided the bases for closing the gap between lab-scale studies and the potential in vivo biomedical application of synthetic motors [14]. Nevertheless, the optimal application in the biomedical field and practical clinical translation of synthetic micro/nanomotors requires understanding of the interactions of these untethered tiny machines with the immune system [15,16]. It may be expected that the entrance of the micro/nanomotors into the bloodstream can lead to undesirable interactions with the immune cells, which in turn may hinder the capacity of the engines to reach their targets and fulfill their task [17]. Therefore, it is necessary to evaluate the biocompatibility of the materials constituting the micro/nanomotors and the energy sources [18]. Furthermore, the physico-chemical properties of the synthetic MNMs should be modulated in such a way that can preclude their clearance from the hosts [19].

This review tries to guide to the reader along the applications of synthetic micro/nanomotors to address some of the current challenges in the drug delivery field, providing an updated perspective on the potential interest in these human-made engines as tools for improving the efficiency of the treatment of different diseases. This requires a careful analysis of the characteristics and properties of this type of system. For this purpose, the review starts with two general sections devoted to the main physico-chemical aspects influencing the fabrication and characteristics of the synthetic MNMs, including their chemical and morphological characteristics, and the effect of the environmental conditions (viscosity of the medium and temperature) on their motion. Then, a detailed discussion of the main mechanisms exploited for guiding the motion of MNMs is included. The last part of the work presents a discussion of some important aspects related to the biosafety of MNMs followed by the introduction of some examples of MNMs exploited for specific drug delivery applications.

## 2. Designing Micro/Nanomotors

The geometry of MNMs is essential for controlling the flow field and pressure distribution during motor propulsion, and hence the optimization of the design of MNMs has driven extensive research activity trying to find the most suitable conditions for the dynamic performance of this type of engine [20]. For instance, the velocity of the motor motion depends on an intricate balance involving the drag forces and the driving forces

acting on the motor, with the increase of the former leading to an increase in the motor speed. On the other side, if both forces reach an equilibrium point, the velocity of the motor motion reaches its maximum speed. Therefore, it is essential to design MNMs with an optimal geometry for optimizing the motion pathway and maximum velocity of the manufactured engines.

### 2.1. Tubular and Rod Motors

Two different geometrical aspects are of a paramount importance in the performance of asymmetric motors: (i) semi-cone angle, and (ii) aspect ratio (ratio between the length and the larger radius) [20]. The former parameter presents importance only in tubular motors, affecting the drag coefficient. For instance, conical shaped motors can reach higher speeds than cylindrical ones with similar aspect ratios [8]. This can be understood as the aspect ratio, which depends on the geometry and chemical properties, of concave surfaces being smaller than that corresponding to convex ones [21]. The importance of the geometrical characteristics of the motor was proven by Wang et al. [22], who showed that the velocity of the engines was significantly increased as the semi-cone angle increases. This is the result of a most favored detachment of the bubbles from concave surfaces in comparison to convex ones. Moreover, the semi-cone angle modifies the aspect ratio, which influences the size of the produced bubbles and their production frequency and enlarges the contact area, favoring the progress of catalytic reactions. This is important because a recent theory suggests that the speed of synthetic motors can be approximately defined by the product of the bubble radius and the generation frequency. Therefore, it is essential to optimize the aspect ratio for controlling the motor speed [23]. According to Li et al. [8], two different regimes may be expected for the dependence of the drag coefficient on the aspect ratio. Thus, when the aspect ratio remains below three, a decrease of the drag coefficient can occur with the increase of the aspect ratio, whereas the drag coefficient increases with the aspect ratio for values of the latter above six. This indicates that the aspect ratio affects the characteristics of the motor motion. In fact, conical motors undergo a faster motion than motors with any other geometry. This can be explained considering the decrease of the drag force acting on the motor as the aspect ratio decreases. Therefore, the propulsion efficiency can be ensured by increasing the semi-cone angle and the aspect ratio in such a way that the drag force operating over the motors can be minimized.

### 2.2. Janus Motors

Janus motors can be prepared with a broad range of structures, including bimetallic structures, shells, and capsules. In the case of bimetallic Janus motors, their motion is commonly the result of a catalytic reaction between the motor surface and the environment. This is the result of the generation of bubbles as a consequence of the catalytic reaction, and their subsequent ejection, which push the motion forward in the motors. The motors can be pulled backward as a result of the instantaneous depression occurring when the bubbles burst [24,25].

The speed of Janus motors can be tuned by modifying their shape. For instance, the use of multilayered Janus hollow capsules can result in a new type of self-propelled engine that can undergo a motion equivalent to 125 times their main dimension per second (about 1 mm/s). Another alternative to increase the velocity of the motor motion is the use of nanoshell motors [21].

### 2.3. Roughness

The origin of the high speed associated with the motion of Janus motors can be found in their inherent surface roughness, which increases the specific area involved in the catalytic process. The work by Orozco et al. [26] reported that the increase of the surface roughness plays an important role in the motion of synthetic motors, controlling the speed of Janus motors, in agreement with the findings of Jurado-Sánchez et al. [27]. They reported that the increase of the roughness of the motors increases the dimension of the catalytic

layer and contributes to an efficient bubble generation and propulsion. The control of the surface roughness as a strategy for modulating the motor speed was also applied to tubular motors. However, the role of the roughness in the motion of the latter is less intuitive because the speed depends on an intricate balance between two forces. The first is the driving force, which contributes to the decomposition of the fuel in the inner region of the motor, and the second is the friction force operating within the outer surface of the motor [28]. This latter contribution is increased with the surface roughness and tends to reduce the speed of the motor motion.

In summary, for Janus motors, the surface roughness can contribute to an effective increase of the speed of the motion. However, the roughness can also introduce undesired friction contributions to the motion, which in turn reduces the speed of the motion.

## 3. Environmental Factors Affecting the Motor Motion

### 3.1. Viscosity

The viscous properties of the fluid surrounding the MNMs play an essential role on their propulsion and speed. For instance, the increase of the environmental viscosity increases the strength of the drag forces operating over the motors, which is opposed to the motion and reduces the motor velocity. Wang et al. [29] reported on the existence of a linear relationship between the velocity of microrockets and the viscosity of the solution. Thus, the increase of the viscosity of the medium reduces the diameter of the produced bubbles and their generation rate, which in turn results in a reduction of the speed of the motion.

The Reynolds number can also affect the motor motion as a result of its dependence on the viscosity. At the highest viscosities (low Reynolds number conditions), the motor motion is commonly linear, whereas it becomes circular as the viscosity decreases and the Reynolds number increases. Moreover, the increase of the viscosity reduces the velocity of the motion due to the increase of the drag force operating over the motor [30].

### 3.2. Temperature

In general, the velocity of the motor increases with temperature, which allows modulating the efficiency of the motor motion. This is, in part, related to the decrease of the solution viscosity as demonstrated by the linearity of the motion at low temperature (higher viscosity) and its circular character at high temperature (lower viscosity) [31].

## 4. Powering the Motion of Micro/Nanomotors

### 4.1. Endogenous Powered Micro/Nanomotors

Endogenous powered self-propelled micro/nanomachines exploit the energy obtained from specific chemical or biochemical reactions for driving their motion [32]. This requires combining two components: a catalytic material (generally a metal or enzyme) and an inert one. Thus, it is possible to fabricate an asymmetric supramolecular architecture that is coated with a specific catalytic material to ensure a continuous energy input from the environment. The asymmetry is essential for the performance of chemically/biochemically propelled MNMs because it ensures the existence of an asymmetry field across the supramolecular system that contributes to power its motion by the generation of local gradients of electrical potential and/or concentration and gas bubbles as the result of surface reactions [33]. Therefore, it is possible to define up to three different mechanisms of propulsion based on chemical reactions. This depends on the mechanism used for the conversion of chemical energy into kinetic energy: self-electrophoresis, self-diffusiophoresis, and bubble propulsion [34]. Self-electrophoresis is associated with an asymmetric production and consumption of ions surrounding the motors. This results in a local electric field guiding the motion. In the case of the self-diffusiophoresis, it is the asymmetry of the chemical species around the motors that is the driving force of the motion. Finally, the diffusion mediated by the generation of gas bubbles is associated with the recoil force of the chemical reactions resulting from the production and growth of bubbles that are separated from the motors, pushing their motion.

A popular alternative is exploiting redox reactions for driving the motion of micro/nanorobots, with the decomposition reaction of hydrogen peroxide being a very frequently approach. This takes advantage of the instability of the hydrogen peroxide, which eases its decomposition into water and oxygen with the participation of several types of catalysts, including metals, enzymes, or alkaline environments. Furthermore, the decomposition of hydrogen peroxide has been widely used as fuel for motors based on bimetallic nanorods, Janus particles, and polymer vesicles [35–37]. These are typical examples of self-electrophoresis–powered motors where the chemical gradient originated as a result of the reaction induces a local electric field, which guides the motor motion. Thus, considering the motion of bimetallic (gold-platinum) Janus nanorods in hydrogen peroxide, a redox reaction of the hydrogen peroxide can be expected, resulting in a production of protons at the platinum end (anode) and their consumption at the gold end (cathode), producing protons. This results in an asymmetric proton distribution within the Janus particle, which induces a local electrical field from the platinum end to the gold one, driving the motion of such electric fields [38].

Solute concentration gradients can be also exploited as an effective method for guiding the micro/nanomotor motion (self-diffusiophoresis). This depends on the nature of the specific solute, which can lead to two different situations: electrolyte diffusiophoresis and non-electrolyte diffusiophoresis [39,40]. Electrolyte diffusiophoresis occurs when the surface reactions produce different anions and cations, which diffuse very differently in the solution. This induces a local electric field that pushes the motion of the motors. In contrast, the non-electrolyte diffusiophoresis relies on the release of non-ionic products in the solution [41].

The propulsion due to the formation of gas bubbles is very common when micro/nanomotors having large catalytic surfaces are considered [20,42]. Thus, it is possible to push the directional motion of the motors as a result of a surface catalytic reaction, which drives the decomposition of the hydrogen peroxide into $O_2$ or $H_2$. In fact, when the concentration of gas accumulated within the motor overcomes its solubility limit, it will overflow as bubbles, pushing the motor motion [26]. It should be noted that the motion speed depends on the concentration of hydrogen peroxide, and hence the higher the hydrogen peroxide concentration, the higher the motion speed. This was demonstrated by Gao et al. [43], who fabricated microtubular engines consisting of a platinum cylinder covered by a poly(aniline) layer 8 µm of length, which can be self-propelled at high speed (around 350 times the microtubule length per second). Furthermore, this type of motor can move even at low concentrations of hydrogen peroxide (0.2% *v/v*). Indeed, the hydrogen peroxide concentration allows modulating the motor speed by tuning the radius of the bubbles and their production frequency. For instance, the increase of the bubble sizes coupled to the decrease of their production frequency leads to a decrease of the speed of the motor motion. On the other hand, the bubble-continuous medium interfacial tension also influences the velocity of the microtubular motors. In fact, the smaller the surface tension, the smaller the bubble size and the higher the production frequency, which in turn enhances the mobility of the motors.

Self-propelled Layer-by-Layer (LbL) Janus capsules obtained by covering with Au and Ni layers the hemispheres of silicon dioxide particles coated with five bilayers formed by the alternate deposition of poly(4-styrenesulfonate of sodium) (PSS) and poly(allylamine hydrochloride) (PAH) have been used for drug delivery applications. For this purpose, catalase was conjugated to the Au layer for driving the catalytic breakdown of hydrogen peroxide to produce gas bubbles that can push the motor motion. The combination of this motion with the magnetic field guidance allows guiding the obtained particles to the target cells, where it is possible to release the encapsulated drugs upon Near Infrared (NIR) irradiation [36]. Figure 1 shows a sketch of the triggered transport and release of doxorubicin (DOX) using LbL Janus capsule motors.

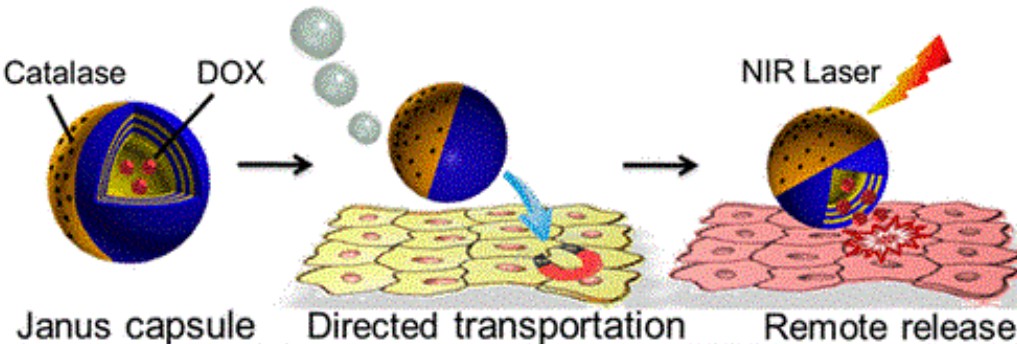

**Figure 1.** Sketch of the triggered transport and release of DOX using LbL Janus capsule-based motors. Reprinted from Wu et al. [36], with permission from the American Chemical Society. Copyright (2014).

It should be stressed that the use of high concentrations of hydrogen peroxide or acid as fuel in chemically driven micro/nanomotors is not suitable when biomedical applications are considered due to the well-known oxidative toxicity of such chemicals [44]. Therefore, it is required to seek more convenient fuels to propel motors when biological media are involved. This can be partially solved by using magnesium-based motors, which present a high biocompatibility and can react with water to produce gas bubbles [6]. This was exploited by Mou et al. [45] for fabricating hemocompatible Janus motors with one hemisphere coated with Pt and a second one with Mg. This type of motor can be propelled by the hydrogen bubbles generated as a result of the reaction between the water and the magnesium, which can lead to the mechanism as summarized in Figure 2.

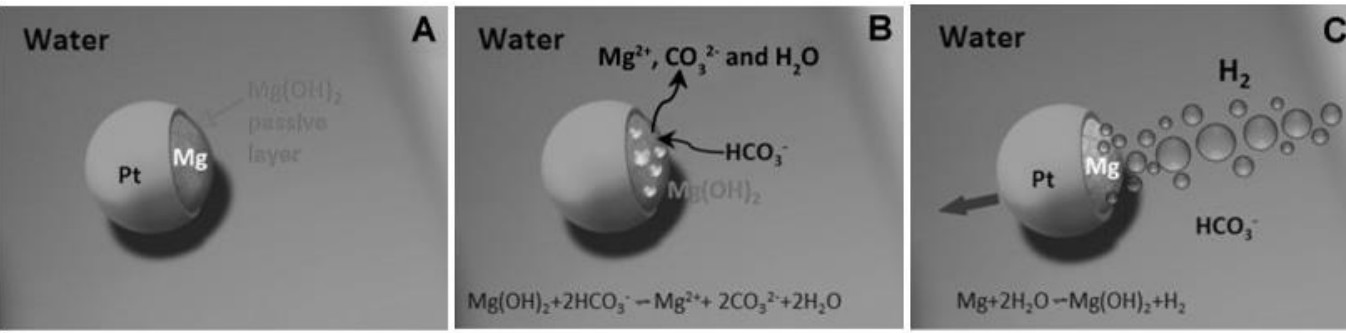

**Figure 2.** Schematic representation of the mechanism driving the motion of Janus motors of Mg and Pt: (**A**) Formation of a passivation layer formed by $Mg(OH)_2$ as a result of the magnesium–water reaction. (**B**) Removal of the $Mg(OH)_2$ passivation layer upon reaction in aqueous media with $NaHCO_3$. (**C**) Release of $H_2$ bubbles from the Mg surface triggered by $NaHCO_3$ to drive the propulsion of the Janus motor. Reprinted from Mou et al. [45], with permission from John Wiley and Sons, Co., Ltd, Hoboken, NJ, USA, Copyright (2013).

The previous example exploits the reaction of water and metals for propelling the motion of micro/nanomotors. This approach, together with photocatalytic water-splitting reactions, can be considered an excellent alternative for pushing the motion of micro/nanomotors by the generation of hydrogen or oxygen [46,47]. This results from active metals, which can react with water smoothly, e.g., magnesium and aluminum. This type of metal can generally form a passivation layer on the surface, reducing the violence of the reaction with water [48]. The seminal work on motors using water as fuel was performed by Gao et al. [49]. They fabricated a Janus microparticle composed of a hemisphere of Ti and a second one having an alloy of Al and Ga. This latter hemisphere in the presence of water reacts, producing hydrogen bubbles that can push the motion of the Janus motor with a remarkable speed of about 3 mm/s (around 150 times the particle diameter per second), exerting a force higher than 500 pN. The high speed of this type of water-driven Janus motor can be in

part ascribed to the large diameter of the generated bubbles (around 10 µm) and the large size of the fabricated motor (average diameter of 20 µm), which ensure a large catalytic surface area and the formation of large bubbles. The propulsion behavior and lifetime of the motors can be tuned by changing the ionic strength or pH of the medium. On the other side, it may be expected that the control of the alloy reactivity by changing the composition or microstructure can contribute to improve the locomotion of the Janus motors. Figure 3 represents the displacement of Janus motors composed of a hemisphere of Ti and a second one of an Al-Ga alloy in water. Wu et al. [50] designed red blood cells coated on one of their sides with an Mg layer that allows an asymmetric generation of hydrogen bubbles, driving the propulsion of the motors without any external fuel and reaching an average velocity of about 172 µm/s.

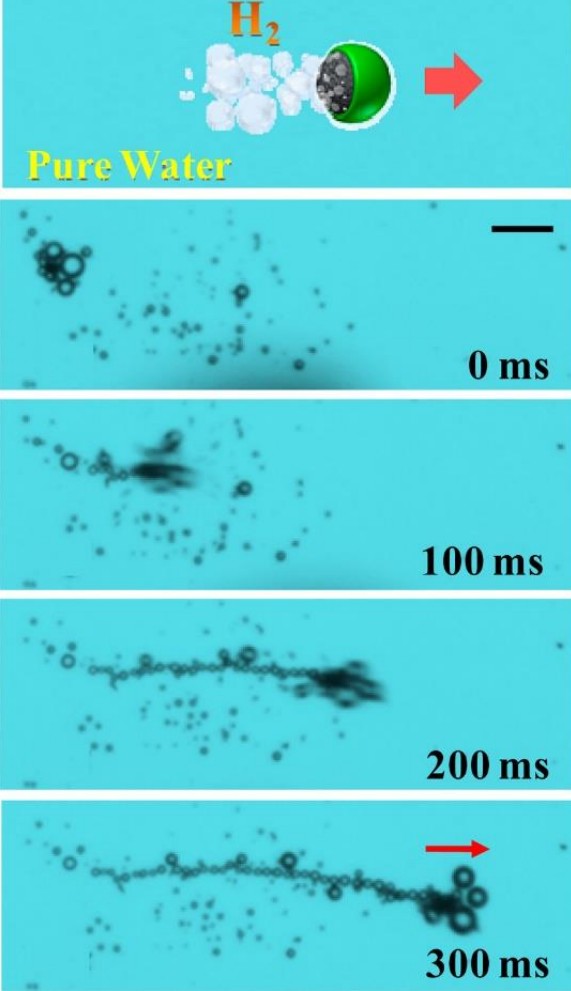

**Figure 3.** Displacement of Janus motors composed of a hemisphere of Ti and a second one of an Al-Ga alloy in water at different times. Reprinted from Gao et al. [49], with permission from the American Chemical Society. Copyright (2012).

It should be stressed that endogenous powered micro/nanomotors are helpful because this type of motor does not require the use of any external stimulus to control the motor during the entire operational time. Unfortunately, they need, in most cases, an external trigger ensuring that the motor is driven to the specific target. On the other hand, the motion of chemically driven motors is not always easy to control and can be easily disturbed, especially in ionic mediums. Furthermore, the use of chemical reactions for powering micro/nanomotors can result in a significant reduction of the power as the reaction approaches the end [16].

Enzyme-Actuated Micro/Nanomotors

Some micro/nanomachines take advantage of specific enzyme-triggered bio-catalytic reactions, which allows the conversion of chemical energy into a mechanical force [51]. This propulsion mechanism based on enzymatic reactions presents as the main advantage the biofriendly character of the used fuels, increasing the value of enzyme-actuated micro/nanomotors (EMNMs) for biomedical applications [52]. Hortelao et al. [53] proposed the design of mesoporous silica-based core-shell motors for the efficient delivery of doxorubicin to cells. For this purpose, they conjugated urease on the surface of the particles. This urea acts as catalyst to stimulate the decomposition of the urea contained in the medium into carbon dioxide and ammonia, which pushes the motion of the motors. This type of propulsion increases significantly (four times) the efficiency of the drug release in comparison with passive systems, resulting in an enhanced anticancer efficiency towards HeLa cells as a result of the synergistic interactions resulting from the combination of the drug release process and the ammonia production due to the catalytic reaction.

Schattling et al. [54] used Janus motors with the enzymatic pair formed by glucose oxidase and catalase conjugated to one of the hemispheres. The catalytic activity of this enzymatic pair was exploited for pushing the motion of the manufactured motor using glucose as fuel. The main problem of this approach is that the enzymatic degradation of the glucose results in the production of $H_2O_2$, and hence it is necessary to ensure that its concentration does not exceed the cytotoxicity threshold. This can be achieved by the action of the catalase on the conversion of the $H_2O_2$ into oxygen and water. Ma et al. [51] reported the first example of Janus nanomotors with dimensions below 100 nm. This type of nanomotor consists of a mesoporous silica nanoparticle coated on one of its sides with a thin silicon dioxide layer, while on the other hemisphere, catalase is bound. Thus, it is possible to push the motion of the motor as a result of the decomposition of $H_2O_2$ triggered by the enzyme catalase. This leads to an enhanced diffusion in comparison to the Brownian diffusion found at low $H_2O_2$ concentrations, opening new avenues for the design of mesoporous motors for active drug delivery. Further studies on the design of mesoporous motor particles were performed by Simmchen et al. [55]. They designed a motor particle where a single-strand DNA was conjugated to one of their faces, and the enzyme catalase to the other one. These motors offer the opportunity to capture and transport different cargos as a result of the presence of specific oligonucleotide sequences that can interact with the single-strand DNA conjugated to the particles.

Abdelmohsen et al. [56] designed bowl-shaped stomatocytes functionalized with the enzymatic pair formed by catalase and glucose oxidase (see Figure 4). This type of nanomotor can move even at low fuel concentrations (hydrogen peroxide or glucose), while the enzyme maintain a high activity. This is possible by the confinement of the enzymes in the designed platform. Therefore, this type of nanomotor combines a high control over the motion and directionality with the protection of the active molecules, offering different opportunities for application, e.g., biosensing, protein and DNA isolation and detection, and immunoassays. It should be noted that enzyme-loaded stomatocytes can be propelled three times faster than stomatocytes based in the reaction of Pt and hydrogen peroxide [57].

It should be noted that the locomotion efficiency of EMNMs can be enhanced by controlling their geometries. In fact, tubular motors present better performance than spherical ones [42].

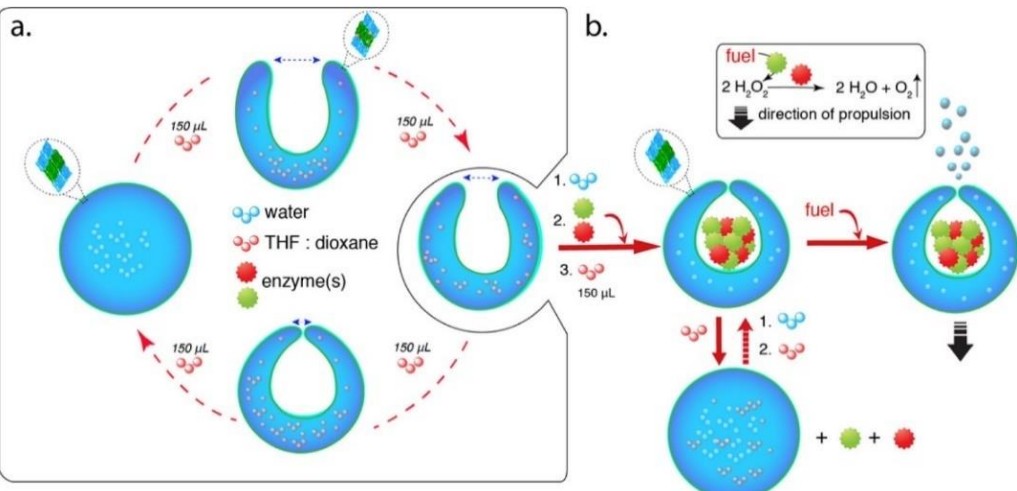

**Figure 4.** (**a**) Schematic representation of the assembly process of active stomatocytes. (**b**) Sketch of the propulsion mechanism of active stomatocytes. Reprinted from Abdelmohsen et al. [56], with permission from the American Chemical Society. Copyright (2016).

### 4.2. Externally Actuated Micro/Nanomotors

The requirement of autonomous motion in the micro/nanoscale in drug delivery applications using synthetic motors or robots may require, in certain cases, driving their motion using an external power input. A broad range of power sources are currently available that can be used as external triggers of the motion of micro/nanomotors in drug delivery applications, e.g., magnetic and electrical fields, light irradiation, acoustic waves, and heat. These can be used independently or in combination to provide multifunctionality to the drug delivery platforms, which allows the exploration of new avenues in the treatment of several diseases [16].

#### 4.2.1. Magnetically Guided Micro/Nanomotors

Magnetic actuation is probably the most promising alternative in the design of self-propelling micro/nanomotors, offering a broad range of swimming strategies, e.g., helical swimmers, flexible swimmers, and surface walkers [58]. Furthermore, the magnetic fields present several advantages in relation to other actuation methodologies, e.g., high penetration, non-invasiveness, and strong controllability. On the other side, magnetic fields can penetrate freely within biological tissues, becoming a very simple operational strategy for driving motor motion [34].

The preparation of magnetically actuated micro/nanomotors commonly requires the presence of ferromagnetic elements (Fe, Co, Ni, etc.), which can be magnetized under the application of an external magnetic field [59]. Thus, it is possible to obtain several types of propulsion depending on the nature of the applied magnetic field: rotating or oscillating magnetic fields. Rotating magnetic fields are characterized by a magnetic induction vector that rotates at a fixed frequency, generating a torque. This drives a forward rotation of the motor, allowing the modification of the motion direction by changing the direction of the applied magnetic field [60]. Oscillating magnetic fields are the result of the combination of a uniform static magnetic field and a rotating one. Thus, it is possible to push a back-and-forth motion of the motor along the direction of the resultant magnetic field, offering long-range driving and navigation capabilities to the manufactured motors [61].

Zhang et al. [60] fabricated artificial bacterial flagella actuated by weak magnetic fields. This type of motor (helical swimmer) consists of a helical tail (InGaAs/GaAs or InGaAs/GaAs/Cr) similar to natural flagellum and a thin magnetic head (Cr/Ni/Au) on one end, allowing a swimming locomotion that can be precisely controlled by three orthogonal electromagnetic coil pairs. In fact, the artificial bacterial flagella work as helical propellers, converting rotary motion to linear motion. The polarity of this motion can

be switched by reversing the rotation direction of the magnetic field. Further studies on the design of motors based on artificial bacterial flagella were performed by Schamel et al. [62]. They designed nano-sized artificial bacterial flagella (around 70 nm), which offer advantages for the control of their motion even through viscoelastic fluids.

A very interesting approach, especially for drug delivery purposes, is the combination of artificial bacterial flagella and drug-loaded liposomes, as demonstrated Qiu et al. [63]. They showed that the application of low-strength rotating magnetic fields to titanium-coated artificial bacterial flagella is an excellent tool for a precise 3D navigation in fluids, allowing the transport of calcein-loaded liposomes deposited on their external surface. This leads to a targeted release of about 73% of the loaded model drug. Following a similar concept, artificial Ni-based bacterial flagella were decorated with lipoplexes containing pDNA, allowing an effective gene delivery to human embryonic kidney cells under in vitro conditions. This is possible by taking advantage of the motion of the magnetic engine under low-strength magnetic fields. Unfortunately, the use of Zn can result in chronic toxicity under in vivo conditions, and it is necessary to carefully analyze the potential applications of this type of system [64].

Medina-Sánchez et al. [65] designed metal-coated polymer microhelices for the capture, transport, and release of immotile sperm cells under conditions mimicking those occurring during the fertilization process. Thus, it is possible to deliver single sperm cells on the oocyte wall under the action of magnetic fields. The use of magnetically driven micromotors was later extended to other aspects related to the reproductive medicine, such as embryo implantation. For this purpose, Schwarz et al. [66] tested two types of magnetic micropropellers, helixes and spirals, for capturing and transporting bovine and murine zygotes, taking advantage of the propulsions induced by the application of a rotating magnetic field. This approach allows propulsion and cargo transportation within high-viscosity mediums and confined microfluidic channels. Moreover, it is possible to transfer cell-loaded motors between different environments, offering new opportunities for in vivo application in embryo transfer.

Flexible swimmers such as Au-Ag-Ni nanowires based on an undulatory locomotion mechanisms are also interesting alternatives in the fabrication of magnetically driven synthetic motors. The motion of this type of engine under the application of a rotating magnetic field occurs by the rotation of the Au and Ni segments at different amplitudes. This is possible because the torque produced by the rotation of the Ni segment as a response to the magnetic stimulus is transmitted along the Ag flexible segment to the gold head forcing its rotation. Therefore, it may be assumed that the motion of this type of swimmer is triggered by the breaking of the system symmetry. The modification of the lengths of the Au and Ni segments and the modulation of the applied magnetic field allow designing engines with forward (pushing) and backward (pulling) magnetically powered motion, and a precise switch "on/off" of the motion, respectively, providing a promising strategy for transport of biological media such as urine [67]. Figure 5 represents a comparison of the mechanism of forward and backward motion.

Jang et al. [68] fabricated magnetically composite multilink nanowire-based chains (diameter 200 nm), which undergo an undulatory motion under the application of a planar-oscillating magnetic field. This type of swimmer can be considered as eukaryote-like systems constituted by a polypyrrole tail and a series of rigid magnetic nickel links that are connected through flexible polymer bilayer hinges.

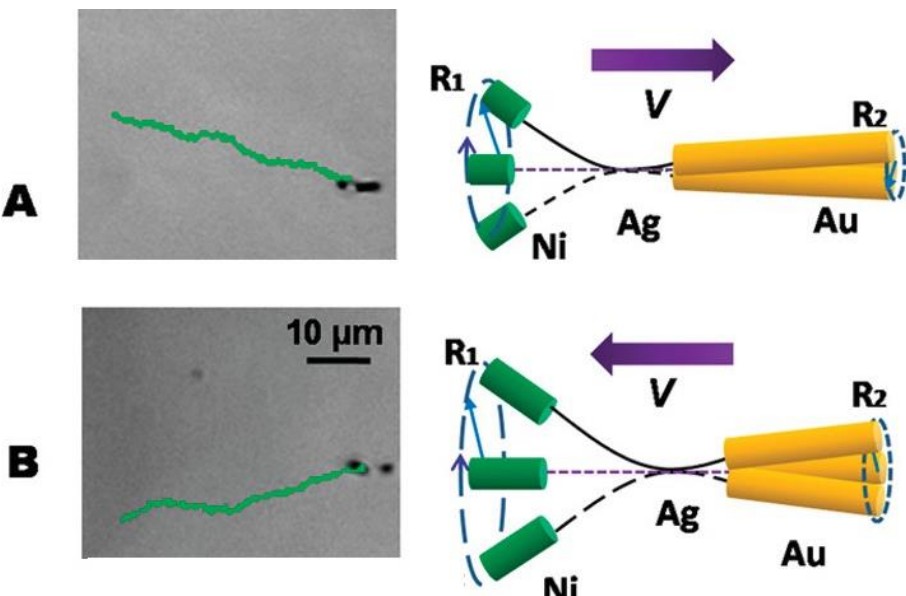

**Figure 5.** Comparison of the forward (**A**) and backward (**B**) motions of Ni-Ag-Au flexible microswimmers. The panel on the left represents the motion of the rods under the application of the magnetic field, and the panel on the right represents the corresponding strategies leading to the microswimmer motion. Adapted from Gao et al. [67], with permission from the American Chemical Society. Copyright (2010).

4.2.2. Electric Propulsion of Micro/Nanomotors

The fabrication of MNMs using conductive materials allows for the exploitation of the conversion of the electric energy into motion through two types of mechanisms: (i) electro-osmotic flow propulsion and (ii) electric current dynamic flow propulsion. Thus, it is possible to tune the motion direction as well as the speed by regulating the surface charge of the motor or the electrochemical reactions occurring at the motor/fluid interface [34]. The propulsion guided by an electroosmotic flow occurs in polarized particles when they accumulate opposite charges in their electrical double layer as a result of the application of an AC electric field. This generates a DC local electric field and the electro-osmotic flow, pushing the motion of the motors as a result of the constant electric field emerging between the electrode and the local electro-osmotic flow [69]. On the other hand, electric current propulsion occurs upon the application of a high-intensity electric field in a medium with low conductivity. Thus, it is possible to generate a current body dynamics that propel the liquid motion, pushing the active translational and rotational motion of the motors [70].

One of the seminal works on electrically driven MNMs was authored by Calvo-Marzal et al. [71]. They proved that the motion of Pt/Au nanowire motors within a hydrogen peroxide medium may be triggered under the application of an external electrical field. This strategy allows a cyclic on/off activation of the motor motion as well as a fine control over the speed. For instance, the decrease of the applied voltage from 1 to 0.4 V leads to an increase of the motor speed from 4 μm/s to 20 μm/s, whereas in the absence of any applied electric field, the motion occurs at 9 μm/s. Fan et al. [72] designed synthetic motors based on the use of gold nanowires decorated with cytokines. These motors can be moved under the application of constant and alternating currents to generate electrophoretic and dielectrophoretic forces. Moreover, the application of the electrical field can contribute to the cell stimulation once the motors stick on their surfaces. It is worth mentioning that the combination of constant and alternating current allows simultaneous control of the motion and positioning directions, allowing a displacement more than two times faster than those obtained with the motors designed by Calvo-Marzal et al. [71], i.e., reaching velocities of up to 50 μm/s.

Rahman et al. [73] demonstrated that carbon nanotubes in aqueous medium can rotate upon the application of a rotating electric field of a specific magnitude and angular speed by taking advantage of the orientations of the water dipole. Thus, a rotational motion at ultrahigh speed (higher than $10^{11}$ rpm) is possible, allowing the rotational transport of attached loads. Guo et al. [74] designed Pt-based bimetallic nanorods with versatility for the transport of cargos to specific targets and the ability to integrate into chemically powered nanomechanical actuators. The combination of AC and DC electric fields allows the alignment and control of the motor, respectively. Moreover, the use of a set of 3D orthogonal microelectrodes for turning on and off the motor motion also allows the motion of the motors within the vertical direction.

It should be stressed that even though synthetic MNMs actuated by external fields are very promising, their applications in drug delivery are limited by the unfriendly character of some of their components in terms of human health [75].

### 4.2.3. Light-Actuated Micro/Nanomotors

Light can be considered among the most ubiquitous energy sources, hence their use for triggering the motion of MNMs emerges as a promising strategy, especially because light can be transmitted wirelessly and remotely. Moreover, the properties of light, e.g., intensity, frequency, polarization, and propagation direction, can be temporally and spatially adjusted, which creates interesting opportunities to control the motion of MNMs. The basic concept for light-driven motion relies on the breaking of the pressure distribution symmetry by the generation of a light-induced asymmetric field within photoactive particles, pushing their motion. This is possible through different strategies [34].

The most common method of light-induced propulsion relies on the combination of two asymmetric gradient fields, which drive the motion following self-electrophoresis, self-diffusion electrophoresis, or self-thermophoresis mechanisms. The first type of gradient presents a chemical origin and is derived from the photocatalytic reactions occurring upon irradiation, resulting in an asymmetric field that propels the motion [76]. The second type of gradient results from the change of the temperature associated with the irradiation, pushing the motion by autothermal migration [77]. The motor motion can be also triggered by using the bubble recoil principle. This relies on the asymmetric distribution of the bubbles produced as a result of photochemical reactions, which induce a bubble concentration gradient that drives the motor motion [78].

Light can also generate surface tension gradients as a result of photochromic reactions that modify the interfacial properties. This forces the fluid to flow from low interfacial tension regions to high interfacial tension ones, pushing the motion of the motors [79–81].

The last, but not the least, application of light for driving the motion of synthetic motors relies on the ability to induce deformations in specific materials, such as crystal elastomers with high elasticity and liquid crystal order. The irradiation of this type of material with the light of a specific wavelength allows inducing their dilational deformation, changing the fluidity of the materials and powering their motion [82,83].

Zhan et al. [84] designed light-driven motors formed by core-shell $Sb_2Se_3/ZnO$ nanomotors, which exploit the ability of the $Sb_2Se_3$ to adsorb light polarized parallel to the nanowires. This leads to a strong dichroic swimming behavior, which presents higher velocity when the incident light is parallelly polarized than when it is perpendicularly polarized. On the other hand, the combination of two cross-aligned dichroic motors can drive the behavior of polarotactic artificial microswimmers, which can move by controlling the direction of polarization of the incident light.

Wang et al. [85] designed micromotors based in glucose-fueled composites formed by cuprous oxide and N-doped carbon nanotubes activated under the action of environmentally friendly visible light. Thus, it is possible to move the manufactured motors with a velocity of up to 18.71 µm/s, which is similar to that obtained for Pt-based catalytic Janus micromotors fueled in $H_2O_2$ medium. Moreover, the velocity of the new type of motor can be regulated by tuning the glucose concentration or the light intensity. On the

other hand, the glucose-fueled composite motors formed by cuprous oxide and N-doped carbon nanotubes can undergo a highly controllable negative phototaxis behavior. Figure 6 represents a sketch of the locomotion mechanisms of motors formed by cuprous oxide and N-doped carbon nanotubes as well as their trajectories in the absence and presence of glucose fuel.

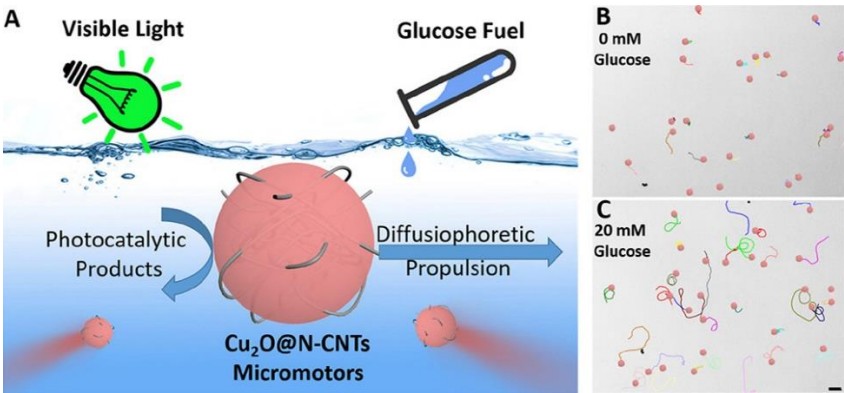

**Figure 6.** (**A**) Sketch of the locomotion mechanism of motors formed by cuprous oxide and N-doped carbon nanotubes. (**B**) Trajectories of motors formed by cuprous oxide and N-doped carbon nanotubes in the absence of glucose fuel. (**C**) Trajectories of motors formed by cuprous oxide and N-doped carbon nanotubes in the presence of glucose fuel. Reprinted from Wang et al. [85], with permission from the American Chemical Society. Copyright (2019).

Xing et al. [86] designed Janus tubular motors constituted of hollow mesoporous carbon particles, with one of the hemispheres coated by a Pt layer. Thus, it was possible to combine a dual actuation mechanism based on the catalytic degradation of the hydrogen peroxide and the irradiation with near infrared light (NIR). The latter provides a directional motion of the fabricated motors as a result of the thermal gradient generated during the irradiation process, favoring adhesion to carcinogenic cells. Similarly, Xuan et al. [87] designed Janus motors based on silica nanoparticles, with one hemisphere coated by a gold layer. This type of system presents a fuel-free motion that is externally actuated by NIR irradiation. This induces a photothermal effect on the Au-coated hemisphere, which leads to thermal gradients across the motor, driving a self-thermophoresis phenomenon. Thus, it is possible to push an ultrafast motion (up to 950 body lengths/s for motors of 50 nm). The use of a remote NIR laser provides a powerful strategy for a reversible "on/off" motion, simultaneously controlling the motion directionality and offering an excellent tool for improving the maneuverability of fuel-free motors. He et al. [88] exploited Janus motor particles guided by the external actuation of magnetic fields or NIR irradiation for inducing a photothermal tissue welding, with results comparable to those expected for common medical sutures. It should be noted that the use of MNMs actuated by NIR irradiation offers several advantages for in vivo applications due to the ability of this radiation to focus on regions of small specific area and to penetrate deeply into the tissues without significant damage [89]. The drawbacks associated with the use of other types of radiation do not preclude the design of motors actuated for them. For instance, the irradiation of AgCl particles with UV light induces a self-diffusiophoretic response due to the asymmetric photodecomposition of the particles, which drives the particle motion in aqueous medium [41].

### 4.2.4. Ultrasound-Actuated Micro/Nanomotors

The use of ultrasound radiation as an external trigger of the motion of synthetic MNMs relies on forcing the particle motion upon the application of acoustic radiation. This can be understood considering that the application of an ultrasonic field to fluids with suspended particles leads to particle–fluid interactions that can induce different motion

states [90]. There are two different mechanisms of ultrasonic propulsion, depending on whether the radiation acts directly on the motor or not. The former is ultrasonic wave propulsion, which requires asymmetric motors and uneven sound pressure distribution, making possible the motion as a result of the pressure gradient [91]. The second mechanism is the so-called acoustic droplet vaporization, which results from the evaporation of liquid droplets. This increases the enthalpy and momentum, generating a powerful pulse to promote the motor motion [92].

The seminal work in ultrasound-powered synthetic motors proved that the application of ultrasonic standing waves in the MHz range can drive the levitation, propulsion, rotation, and assembly in metallic microrods (2 μm long and 330 nm diameter) in aqueous medium [91]. This occurs through an acoustophoretic mechanism resulting from the microrod asymmetry, which drives its axial propulsion. The asymmetry associated with the convex and concave regions leads to an anisotropic contribution of the ultrasound pressure, which results in a pressure gradient at the microrod surface. Thus, the motors are unidirectionally propelled. On the other hand, the metallic rods can align and self-assemble to form spinning chains, which in the case of Janus microrods occurs following a head-to-tail alternating structure. Moreover, the chains can form ring or streak patterns in the levitation plane, with such patterns having a characteristic distance that is about a half of the wavelength of the ultrasonic excitation radiation. Figure 7 shows the different possible motion pathways of ultrasonic-powered Au/Ru microrods. It should be noted that the asymmetry of the microrods plays a central role in the control of their motion. For instance, the higher the asymmetry, the higher the motion speed, as was proven by Garcia-Gradilla et al. [93], who designed multifunctional rod-like nanomotors formed by three different segments (Au-Ni-Au). This type of motor can be propelled by ultrasound radiation and magnetic fields. The interaction of the latter with the Ni segments can be exploited for providing a predefined and controlled motion to the manufactured motors. Moreover, the Ni segments can be exploited for picking up and transporting magnetic materials.

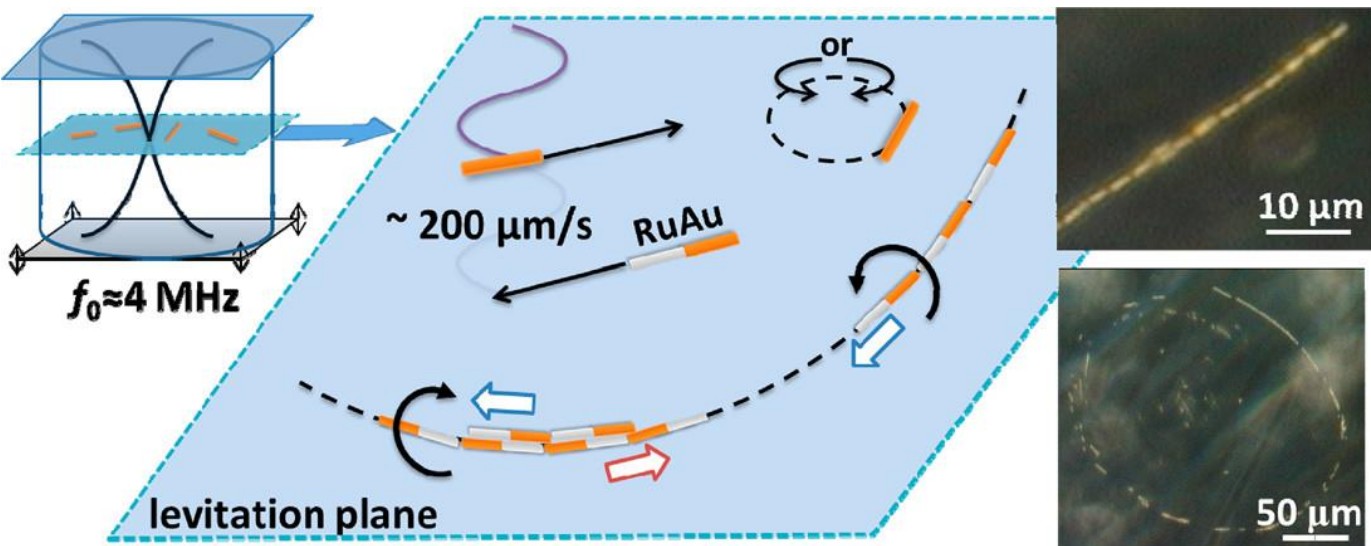

**Figure 7.** Different motion pathways of Au/Ru microrods powered by ultrasonic standing waves. Reprinted from Wang et al. [56], with permission from the American Chemical Society. Copyright (2012).

Tubular motors also offer interesting opportunities for ultrasound-triggered motion, as was demonstrated by Kagan et al. [92]. They showed that ultrasound-triggered vaporization of electrostatically bonded perfluorocarbon droplets contained inside the motor can propel the tubular engines with speeds of up to 6 m/s (two orders of magnitude faster than previous systems). It should be noted that the mechanisms driving the motion of tubular motors present higher efficiency than those occurring in rod-shape ones [34].

Xu et al. [94] showed that the application of an ultrasound field can help in the control of the motion of tubular catalytic motors propelled by hydrogen peroxide. This is possible because ultrasound radiation can disrupt the normal evolution and ejection of the generated bubbles, which is essential to the propulsion of the manufactured motors. Therefore, ultrasound radiation can be used for a precise control of the velocity of the motors by increasing and decreasing sharply the speed of the engine at low and high powers, respectively. For instance, the application of an ultrasound field of up to 10 V reduces the velocity of the manufactured motors from 231 µm/s to 6 µm/s in less than 0.1 s, undergoing a fast recovery upon removal of the applied field. Therefore, this strategy allows extremely fast changes in the motion speed and reproducible "on/off" reversible activation of the motor, improving the efficiency of the energy conversion. Figure 8 shows the ultrasound-modulated motion of chemically powered motors.

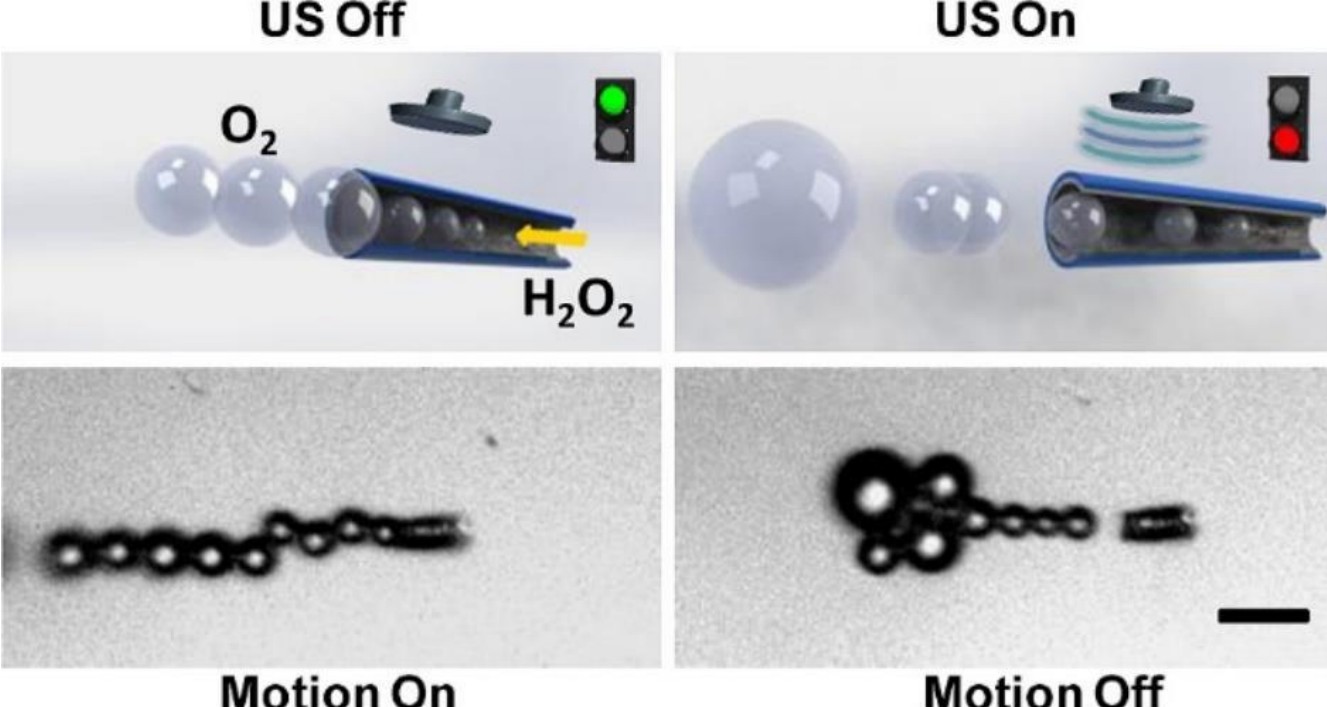

**Figure 8.** Set images (schemes, **top**, and true micrograph, **bottom**) showing the ultrasound-modulated motion of chemically powered motors. Reprinted from Xu et al. [94], with permission from the American Chemical Society. Copyright (2014).

Table 1 presents a comparison between the characteristics of chemically powered and externally actuated motors.

**Table 1.** Comparison of the main characteristics of chemically powered and externally actuated motors. Reprinted from Hu et al. [16], with permission under open access CC BY 4.0 license, https://creativecommons.org/licenses/by/4.0/ (accessed on 4 July 2022).

| Type | Energy | Penetration | Motion Ability | Persistence | Safety |
|---|---|---|---|---|---|
| Endogenous powered motors | Chemical | Not applicable | Requires external force for positioning | Not as good, chemical energy can be depleted when it decreases gradually, limiting the motion of the engines | Depends on the fuel: hydrogen peroxide is a toxic fuel, whereas glucose and urea are safe |
| Exogenous powered motors (Externally triggered) | Magnetic | Good, weak magnetic fields can be enough | Precise 3D navigation in fluids under the action of rotating magnetic fields | Good, engines can keep moving under the guidance of the external field | Used magnetic fields are generally safe, metallic components can present toxicity upon long-term exposure |
| | Electric | Weak, strong electric fields are needed | Requires the combination of electric fields and additional fields for ensuring the directional motion | | Strong electric field can affect human body, metallic components can present toxicity upon long-term exposure |
| | Light | Depends on the type of light, different penetration | Normally exploited for triggering other reactions. However, it can provide directional motion | | Depend on the type of light, ultraviolet light may be harmful, whereas other lights are commonly safe |
| | Ultrasound | Good | Commonly combined with magnetic fields, provides directional motion | | Ultrasound irradiation can cause oxidative stress in cells, metallic components can present toxicity upon long-term exposure |

## 5. Towards the Biocompatibility of Micro/Nanomotors

One of the main challenges associated with the use of micro/nanomotors in biomedicine is the introduction of biocompatible and biodegradable components for the fabrication of miniaturized devices. This is important when biomedical applications of MNMs are considered because of the use of the manufactured motors in biological environments, including cells and tissue [95]. Therefore, the contact between MNMs and the human environment requires analysis of a broad range of factors, including protein adhesion, stimulation of the immune response, biodistribution, toxicity, degradation, and elimination profiles [18]. These factors may influence the activity of MNMs and reduce their effectiveness.

Synthetic MNMs can be manufactured almost at will to provide a suitable response to a specific demand. However, they present different concerns associated with their biological safety under in vivo conditions. This has driven important research activity towards the fabrication of biological motors using natural cells. For instance, different self-driving biological motors have been fabricated using motile bacteria, neutrophils, sperm cells, and

cardiomyocytes [96–99]. These MNMs are characterized by a good compatibility without causing any adverse immune response. However, the number of cells that can be used as substrates for the fabrication of MNMs is limited, and the size of these motors is limited to the micrometric scale. On the other hand, the effectiveness of biological motors in the treatment of different diseases should be carefully examined. The situation changes when truly synthetic MNMs are considered. These can be manufactured with a broad range of shapes and sizes (from a few nanometers to several millimeters). On the other hand, there is a growing interest on the fabrication of synthetic motors following a bionics approach, i.e., by combining biological and synthetic components. This allows combining the biocompatibility of biological blocks and the modularity of synthetic ones, which can create promising opportunities for specific applications [50]. An example of this type of approach is the engineering modification of natural cells by physical or chemical methods, which provides the basis for the introduction of new functionalities to the cells, respecting their biocompatibility. For instance, to ensure the locomotion of this new motor, it is common to include magnetic materials, e.g., iron oxides or magnesium, or enzymes that catalyze specific chemical reactions in aqueous medium or urea [95].

Chemical and physical MNMs can be designed using degradable or self-destructive materials, which can be destructed once they have completed a specific task. However, there are many concerns about the final fate of biological motors after they complete their life cycle. In some cases, they can follow their own metabolic cycles without any safety concerns [100]. Unfortunately, motors having self-proliferation capabilities, e.g., bacteria, should be analyzed more carefully. In particular, it is necessary to design them in such a way that they cannot move to undesired sites where they can proliferate. This requires specific strategies, as proposed by Stanton et al. [101], who used a the local $NH_3$ concentration resulting from urea hydrolysis to stop the motion of bacterial motors on demand by killing the biological activity of the bacteria.

The application of MNMs also requires ensuring the biosafety of the power sources [95]. This is of a paramount importance because a broad number of works have dealt with the use of catalytic motors based on noble metals or metal oxides characterized by a motion ability triggered by the degradation of the hydrogen peroxide, which is toxic to the human body [26,43]. A common alternative for reducing the potential harmful effects of the toxic fuels is the reduction of the concentration of specific molecules, e.g., hydrogen peroxide or mineral acids, in the fuel used for the motor motion. Unfortunately, this reduces the efficiency and speed of the motor motion [95]. The drawbacks associated with the use of hydrogen peroxide as fuel have driven an important piece of research towards the use of enzyme-catalyzed reactions for powering motors. Some examples include urease, which can convert urea to $NH_3$ and $CO_2$; catalase, which converts the hydrogen peroxide in water and oxygen; and glucose oxidase, which catalyzes the conversion of glucose in gluconic acid and hydrogen peroxide [21,102]. This type of system is important because it allows extending the number of driving systems available for MNMs, reducing the possible toxicity. However, they require in most cases higher fuel concentration than that corresponding to the physiological conditions.

In recent years, MNMs using water as fuel have been designed, taking advantage of the reaction of water with metals, e.g., magnesium, aluminum, or other reactive metals, to produce hydrogen, which is theoretically possible. However, the true situation is far from satisfactory due to the formation of passivation layers on the metal surface that reduce the efficiency of the reaction for powering the motor motion, requiring the additional chemicals, e.g., sodium bicarbonate, to the environment and limiting the use of this type of motor to non-basic media, e.g., the acidic environment of the human stomach [95].

In the case of physical motors, it was previously stated that their motion is driven by different external fields without the addition of any fuel [103], and hence the biosafety of this type of system depends on whether the intensity and time of the applied stimulus can cause any damage to normal tissues.

Biological motors that present the capability of autonomous motion, e.g., sperm cells, can move within the human body without adding any fuel or using additional devices. Therefore, their biosafety is almost ensured [104]. However, the biosafety of composite motors containing biological and synthetic pieces depends on the specific characteristics of the driving system [95].

## 6. Micro-/Nano-Motors in Drug Delivery

The exploitation of human-made micro/nanomotors to deliver therapeutic drugs to specific targets represents a novel approach for the treatment of different diseases [5]. This is possible because micro/nanomotors offer different advantages in comparison to more classical drug vectors. These advantages include a rapid drug transport, high tissue penetration, and controllable motion [6]. Indeed, the autonomous motion of micro/nanomotors provides the bases for the controlled transport of drugs to reach tissues that are difficult to access [16]. The fabrication of micro/nanomotors for drug delivery is commonly based on the combination of an internal payload with an external shell, which contributes to the active transport of the device to reach specific targets [105]. Many drugs, including small molecules, small interfering ribonucleic acid (siRNA), DNA, peptides, antibodies, and proteins can be delivered from MNMs [106].

The design of MNMs for drug delivery applications requires consideration of the material used for the fabrication, the cargo, and the mechanism used for guiding the motion. This is important because they affect the different characteristics of the final products, including their size, shape, charge, and the fate of MNMs as defined by their tissue accumulation, intracellular transport, biodegradability, or biocompatibility [107].

The effectiveness of MNMs for drug delivery within the gastrointestinal tract was proven by Esteban-Fernández de Ávila et al. [108], who used Mg-based motors loaded with clarithromycin to treat mice infected with *Helicobacter pylori*. The fabricated motors were administrated orally, evidencing a good ability to be propelled within the gastric fluid. Moreover, the administration of these engines contributed to a reduction of almost two orders of magnitude of the population of *Helicobacter pylori*, without significant toxicity for the mice. A similar approach was followed by Gao et al. [109], who fabricated Zn-based motors. These motors undergo an acid-driven propulsion in the stomach, presenting an effective binding and retention in the stomach as well as good cargo payloads on the stomach walls. Moreover, the motors can be progressively dissolved in the gastric acid, which leads to an autonomous release of the encapsulated drugs without any evidence of toxicity.

Baylis et al. [110] used gas-generating particles formed by the combination of carbonate and tranexamic acid for halting hemorrhage through blood vessels in mice and pigs. This is possible by the locomotion of the particles loaded with thrombin at velocities of up to 1.5 cm/s through aqueous medium, mimicking the main characteristics of the blood. Thus, the combination of different mechanisms, including lateral propulsion, buoyant rise, and convection, push the motors through the fluid, allowing the use of this type of system as an effective hemostatic agent, making possible the halting of hemorrhages in several animal models of intraoperative and traumatic bleeding.

Kim et al. [111] designed magnetically actuated hydrogel engines for the transport and subsequent release of (poly-D,L-lactic-co-glycolic acid particles) loaded with doxorubicin into the eyes. The particularity of this type of engine is its ability to remove the magnetic components from the eyes after drug delivery, which minimizes the possible side effects. Thus, the application of alternating magnetic fields at the target point leads to the dissolution of the therapeutic layers, resulting in the release of the drug; then, the magnetic components are retrieved again under the application of a new magnetic field. Moreover, ex vivo and in vitro studies showed the ability of this type of engine to migrate towards the vitreous, which enables a significant therapeutic effect against retinoblastoma cancer cells.

Cancer induces strong oxidative stress in cells, which leads to the production of high amounts of $H_2O_2$. This can be exploited as an energy source for driving the motion of drug

carriers [112]. Villa et al. [113] designed superparamagnetic/catalytic robots consisting of Janus micromotors formed by an iron oxide particle decorated with tosyl groups, with one of its hemispheres coated by a platinum layer (see Figure 9). This provides a multifunctional character to the motor: (i) the tosyl group layer provides the capacity for binding molecules and biological materials; (ii) the Pt layer contributes to the catalytic decomposition of hydrogen peroxide, helping in the propulsion of the motor; and (iii) the magnetic particle allows manipulation under the application of magnetic fields. In fact, this latter part makes it possible for the motor to work as a single unit or assembly of chains for performing collective actions (e.g., capture and transport of cancer cells). These motors can be exploited for the release of anticancer drugs, e.g., doxorubicin, showing a significant reduction in the proliferation of carcinogenic cells.

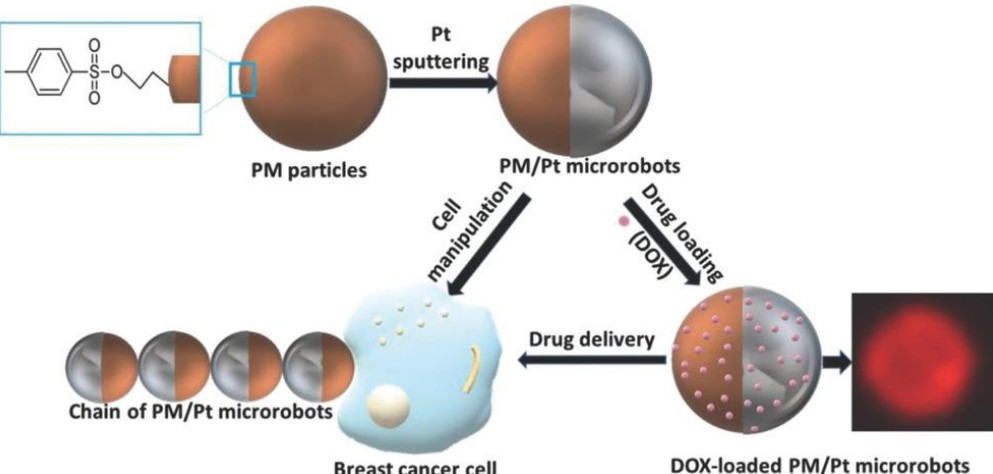

**Figure 9.** Sketch of the fabrication process and action mechanism of catalytic Janus motors loaded with the anticancer drug doxorubicin (DOX). Reprinted from Villa et al. [113], with permission from John Wiley and Sons, Co., Ltd, Hoboken, NJ, USA, Copyright (2018).

Kagan et al. [35] designed very complex $Ni/(Au_{50}/Ag_{50})/Ni/Pt)$ nanowire motors with the ability of picking up, transporting, and releasing doxorubicin encapsulated in biodegradable particles (poly-D,L-lactic-co-glycolic acid particles) and liposomes doped with $Fe_3O_4$ (sizes in the range 100 nm–3 μm) to specific targets. These catalytic motors combine two actuation mechanisms for ensuring a fast propulsion and a directional motion. In fact, they are propelled by the decomposition of hydrogen peroxide, whereas the application of a magnetic field ensures their directional guiding, with a velocity three times faster than that expected for passive motors. Gao et al. [114] designed fuel-free motors based on magnetically actuated flexible nickel-silver swimmers (5–6 μm in length and 200 nm in diameter). This type of engine allows the transport of doxorubicin to HeLa cells at high speeds (more than 10 μm/s, which is equivalent to more than 0.2 body lengths per revolution in dimensionless speed), following a process including several steps: capture drug-loaded magnetic polymeric particles, transport the particles through the channel, approach and stick onto the HeLa cells, and release the doxorubicin. However, this type of motor can be affected by a poor adhesion between the particles and the motors, which can lead to the failure of the motor system before reaching its target. This can be avoided by coating the drug-loaded particles with a polymer layer, which can contribute to the drug–cell membrane binding process.

Manganese oxide–based ($PEDOT/MnO_2$) catalytic tubular micromotors were evaluated as a tool for delivery of the chemotherapeutic drug camptothecin. This is possible by the catalytic decomposition of hydrogen peroxide, which propels an effective autonomous motion in biological media with high speed (318.80 μm/s). One of the main advantages of this type of motor is related to its capability of operating at low fuel concentrations (below 0.4% *w/w*) [115]. Wu et al. [116] fabricated Pt nanorockets with a biocompatible coating

formed by a Layer-by-Layer film of chitosan and alginate for the transport of doxorubicin. This type of engine allows a targeted transport of the drug following a propulsion mechanism driven by the catalytic degradation of hydrogen peroxide, and a controlled release of the drug. In fact, the catalytic degradation of the hydrogen peroxide by the internal core of the nanorockets releases a tail of oxygen bubbles, propelling the engines at a 74 µm/s. The polydispersity of the fabricated engines and the distribution of the Pt nanoparticles within the nanorockets results in different motion pathways, including straight, circular, curved, and self-rotating motions. On the other hand, the application of an ultrasound field was used for triggering the rupture of the capsules, ensuring the release of the drug.

Xuan et al. [117] fabricated a self-propelled Janus nanomotor (diameter about 75 nm) for the transport and controlled release of doxorubicin encapsulated within liposomes on cells. These motors were based on mesoporous silica nanoparticles with caps of chromium and platinum, using the bubble generated as a result of the catalytic decomposition of hydrogen peroxide as the driving force of the motor motion, which can occur at speeds of up to 20.2 µm/s. The study of the in vitro intracellular localization evidences that the fabricated motors can enter into the cells, and the release of the encapsulated drug occurs by the decomposition of the liposomes within the intracellular region. Figure 10 displays the fabrication process and performance mechanism of the Janus motors.

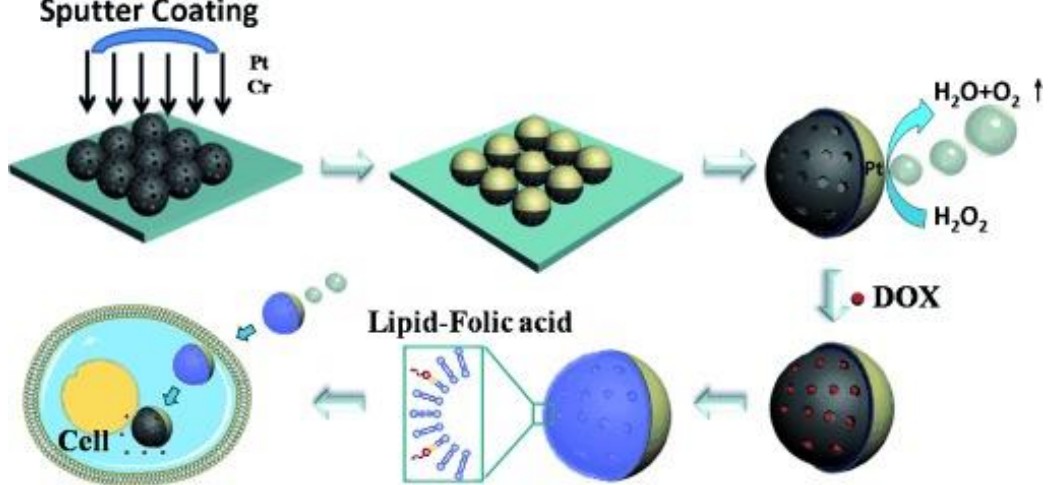

**Figure 10.** Sketch of the fabrication process and action mechanism of catalytic Janus motors on the transport of liposomes loaded with the anticancer drug doxorubicin. Reprinted from Xuan et al. [117], with permission from John Wiley and Sons, Co., Ltd, Hoboken, NJ, USA, Copyright (2014).

Hortelao et al. [53] fabricated urease-powered mesoporous silica-based core-shell motors for the loading, transport, and efficient release of doxorubicin to cells. This is possible due to their ability to undergo self-propulsion in ionic media. This allows a release of the encapsulated drug four times faster than that resulting from passive systems. Thus, the efficiency of the anticancer drug towards HeLa cells is enhanced due a synergistic action of the drug release and the ammonia produced by the degradation of the urea in the medium. The high efficiency of this type of motor may create new opportunities for biomedical applications.

Kehzri et al. [118] fabricated high-speed tubular electrically conductive engines by combining reduced graphene oxide as a platform for an effective drug delivery and platinum as a catalytic core. This type of machine can be loaded with doxorubicin, which can be expelled during motion due to the application of an electron current, resulting in a high therapeutic efficiency against cancer cells, with a significant reduction of the side effects towards healthy tissues. Therefore, this type of motor provides an important step forward in the fabrication of advanced drug delivery systems.

Table 2 summarizes some examples of synthetic MNMs exploited for drug delivery purposes and their power source.

**Table 2.** Examples of synthetic MNMs used for drug delivery purposes.

| Type of MNM | Power Source | Disease | Drug | Reference |
|---|---|---|---|---|
| Mg-based motors | Chemical (catalytic motors powered by gastric acids) | gastrointestinal bacteria | clarithromycin | Esteban-Fernández de Ávila et al. [108] |
| Zn-based motors | | | | Gao et al. [109] |
| Carbonate and tranexamic acid particles | Chemical | hemorrhages | thrombin | Baylis et al. [110] |
| Hydrogel/magnetic particle hybrid | Magnetic fields | eye diseases | doxorubicin | Kim et al. [111] |
| Iron oxide particles with one hemisphere coated by Pt and decorated with tosylated groups | Chemical (catalytic motors powered by decomposition of hydrogen peroxide) | cancer | doxorubicin | Villa et al. [113] |
| Ni/(Au$_{50}$/Ag$_{50}$)/Ni/Pt nanowires and Fe$_3$O$_4$ particles | Chemical (catalytic motors powered by decomposition of hydrogen peroxide) combined with magnetic field (directionality control) | | | Kagan et al. [35] |
| Pt nanorockets coated by a Layer-by-Layer film of chitosan and alginate (tubular motors) | Chemical (catalytic motors powered by decomposition of hydrogen peroxide) | | | Wu et al. [116] |
| Janus nanomotors with caps of chromium and platinum | | | | Xuan et al. [117] |
| Silica-based nanoparticles decorated with urease | Chemical (enzymatic degradation of urea by urease) | | | Hortelao et al. [53] |
| Flexible nickel-silver swimmers | Magnetic field | | | Gao et al. [114] |
| Carbon-platinum tubular Janus motors | Chemical (catalytic motors powered by decomposition of hydrogen peroxide) and light (near infrared radiation) | | | Xing et al. [86] |
| Tubular motors of platinum and reduced graphene oxide | Electric field | | | Kehzri et al. [118] |
| Tubular (PEDOT/MnO$_2$) micromotors | Chemical (catalytic motors powered by decomposition of hydrogen peroxide) | | camptothecin | Feng et al. [115] |

## 7. Concluding Remarks

Synthetic micro/nanomotors (MNMs) are at the forefront of the nanomedical tools designed for improving the diagnosis and treatment of a broad range of diseases. However, while important progress has been made for ensuring the efficient motion of these types of engine, the true application of these small-scale devices is still in its infancy, presenting different problems and challenges. For instance, the biosafety of power sources and fuels as well as that of the materials used for engine fabrication must be taken into consideration

to reduce the risks and hazards associated with their use in the human body. Moreover, the capability of targeting the motors also is of paramount importance in their design. However, in most cases there is a poor understanding of the true framework involving the performance of this type of engine, which is in part the result of the poor understanding of their behavior under in vivo conditions. The understanding of the in vivo performance of MNMs is the only way to verify the effect of their interactions with complex body fluid environments as well as the effects associated with the side effects of long-term persistence of motors within the human body. Therefore, the fabrication of ideal MNMs requires consideration of their capabilities of precise targeted motion and autonomous drug delivery, without compromising their biosafety. The field is open to research and innovation for building safe and efficient engines for drug delivery. In fact, the incorporation of synthetic MNMs as a true therapeutic option for the treatment and prevention of different diseases is far from reality, and additional research on materials and fuels as well as efficiency tests under in vivo relevant conditions are required.

**Author Contributions:** E.G., conducted the review and wrote the draft; E.G. and A.M. revised the manuscript and contributed to substantial enhancement of the manuscript. All authors have read and agreed to the published version of the manuscript.

**Funding:** This work was funded in part by MICINN under Grant PID2019-106557GB-C21 and by E.U. on the framework of the European Innovative Training Network—Marie Sklodowska-Curie Action Nano Paint (Grant Agreement 955612).

**Institutional Review Board Statement:** Not applicable.

**Informed Consent Statement:** Not applicable.

**Data Availability Statement:** Not applicable.

**Conflicts of Interest:** The authors declare no conflict of interest. The funders had no role in the design of the study; in the collection, analyses, or interpretation of data; in the writing of the manuscript, or in the decision to publish the results.

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
