# Peer review of "Synthetic Micro/Nanomotors for Drug Delivery"

_technologies, doi:10.3390/technologies10040096_

Round 1
Reviewer 1 Report
The review manuscript is devoted to a valuable subject. It is illustrated by nice Figures.
Suggestions and critical comments. I suggest to revise the manuscript. It would be useful to see in the revised manuscript a brief description of preparation of a MNM. It is difficult to understand from the Table 1: which motors are externally actuated, externally driven by one or another type of action. How motors can be endogenous powered by light, ultrasound?
Why do authors refer mainly to Reynolds numbers of the surrounding media and not to a more common dynamic viscosity?
Grammar of the manuscript is very bad.
See, e.g. lines 20-21, 38, 49 (Reynolds), 49 Reynolds number, overcoming the Browning motion (?); 240, 241, etc., etc.
Author Response
The review manuscript is devoted to a valuable subject. It is illustrated by nice Figures.
Suggestions and critical comments. I suggest to revise the manuscript.
We have revised the manuscript taking into consideration the point raised by the reviewer.
It would be useful to see in the revised manuscript a brief description of preparation of a MNM.
The point of the reviewer is very interesting. Unfortunately, it means to write almost a new review dealing only with the preparation of micro/nanomotors because this type of systems is a very heterogeneous family containing different types of engines with different chemical nature and structure. Therefore, it is not possible to afford the description of a micro/nanomotor as suggests the reviewer.
It is difficult to understand from the Table 1: which motors are externally actuated, externally driven by one or another type of action. How motors can be endogenous powered by light, ultrasound?
We agree with the reviewer. In table I, there was a notation problem that has been solved in the revised version of the manuscript.
Why do authors refer mainly to Reynolds numbers of the surrounding media and not to a more common dynamic viscosity?
We have rewritten the text to a better understanding of the concepts.
Grammar of the manuscript is very bad.
See, e.g. lines 20-21, 38, 49 (Reynolds), 49 Reynolds number, overcoming the Browning motion (?); 240, 241, etc., etc.
We have revised the whole manuscript for ensuring the correctness of the text.
We thank to the reviewer for the comments, they were very useful for improving our manuscript.
Reviewer 2 Report
Paper deals with important task. The authors conducted an overview of the potential applications of micro/nanomotors in drug delivery, without forgetting the most fundamental aspects related to their performance and biosafety.
Paper has great practical value.
Suggestions:
1. The abstract section should be extended using details fo the results obtained in this paper
2. The introduction section should be extended using more clearly motivation of this paper.
3. It would be good to add point-by-point the main contributions at the end of the Introduction section
4. It would be good to add the remainder of this paper
5. The methodology is unclear. Please provide information about it
6. The conclusion section should be extended using: 1) clear results obtained in the paper; 2) limitations of the conducted review; 3) prospects for future research.
7. Some of references are outdated. Please fix it using 3-5 years old papers in high-impact journals.
Author Response
Paper deals with important task. The authors conducted an overview of the potential applications of micro/nanomotors in drug delivery, without forgetting the most fundamental aspects related to their performance and biosafety.
Paper has great practical value.
Suggestions:
- The abstract section should be extended using details for the results obtained in this paper
The current paper is a review where an updated perspective of a specific topic is presented, and this has been stated in the abstract. It cannot be included more results.
- The introduction section should be extended using more clearly motivation of this paper.
We have tried to provide a better motivation for the manuscript.
- It would be good to add point-by-point the main contributions at the end of the Introduction section
We have included the different aspects analyzed in the work at the end of the introduction.
- It would be good to add the remainder of this paper
What does the reviewer mean with remainder?
- The methodology is unclear. Please provide information about it
What does the reviewer mean with methodology? Our work is a review, and the methodology has been a detailed bibliographic research. However, we think that this point should not included in the main text because it is easy to understand.
- The conclusion section should be extended using: 1) clear results obtained in the paper; 2) limitations of the conducted review; 3) prospects for future research.
We have revised the conclusions.
- Some of references are outdated. Please fix it using 3-5 years old papers in high-impact journal
We cannot agree with the reviewer comment, the review contains more than 25% of the references from articles of the last 5 years, and more than 75% of the references have less than 10 years. Moreover, these references are of papers in journal of very high impact on the field. Therefore, we consider that the review provides a very suitable representation of the current state-of-the-art of the discussed topic.
We thank to the reviewer for the comments, they were very useful for improving our manuscript.
Reviewer 3 Report
The presented manuscript is a review focussed on the design, properties and application of micro/nanomotors and especially their biomedical applications. Those kind of structures have unique properties and can allow to develop different smart delivery system moving because of energy transformation.
The topic of the study is relevant to the Journal. The manuscript structure is in correspondence to the Journal requirements. Therefore I recommend its publications in Technologies but after minor revision.
I have recommendations and comments:
-
The properties of MNMs are described very well, but my opinion is that in the introduction section will be better to have more examples of specific particles.
-
The particles can be produced with size from nm to millimeters, but would you like to dive more information or correlation in the influence of size and charge on the biomedical application of the particles (the motion of MNMs will be strongly affected from their charge and geometry?)
Author Response
Reviewer 3:
The presented manuscript is a review focused on the design, properties and application of micro/nanomotors and especially their biomedical applications. Those kind of structures have unique properties and can allow to develop different smart delivery system moving because of energy transformation.
The topic of the study is relevant to the Journal. The manuscript structure is in correspondence to the Journal requirements. Therefore I recommend its publications in Technologies but after minor revision.
I have recommendations and comments:
- The properties of MNMs are described very well, but my opinion is that in the introduction section will be better to have more examples of specific particles.
The point raised by the reviewer will destroy the message of the introduction, and we prefer to maintain the mentioned discussion as in the present form.
- The particles can be produced with size from nm to millimeters, but would you like to dive more information or correlation in the influence of size and charge on the biomedical application of the particles (the motion of MNMs will be strongly affected from their charge and geometry?)
The point raised for the reviewer is very interesting, unfortunately up to date there are no systematic information about such aspects, and hence it is not possible to include a sound discussion.
We thank to the reviewer for the comments, they were very useful for improving our work.
Reviewer 4 Report
This review is dealing with synthetic micro/nanomotors (MNMs) characterized by their capacity for undergoing self-propelled motion as result of the consumption of chemical energy or external physical stimulus.
This review further provides an updated perspective to the potential applications of synthetic MNMs in drug delivery with particular discussion of their biosafety.
This review is well organized and referred suitably previous related papers and thus will be published without further correction.
Author Response
Reviewer 4:
This review is dealing with synthetic micro/nanomotors (MNMs) characterized by their capacity for undergoing self-propelled motion as result of the consumption of chemical energy or external physical stimulus.
This review further provides an updated perspective to the potential applications of synthetic MNMs in drug delivery with particular discussion of their biosafety.
This review is well organized and referred suitably previous related papers and thus will be published without further correction.
We thank to the reviewer for the comments, they were very useful for improving our work.
Round 2
Reviewer 1 Report
The revised version is OK. Minor English correction by a technical editor is desirable.
Author Response
Response to the comments of the reviewers
Reviewer 1:
The revised version is OK. Minor English correction by a technical editor is desirable.
We have checked the new version of the manuscript for language correctness.
We thank to the reviewer for the comments, they were very useful for improving our work.
Reviewer 2 Report
Dear authors,
thank you for the improvement of your paper.
1. Remainder - structure of the paper. This information should be added at the end of the Introduction section.
2. Modern papers with Literature review always used some methodology that help to reproduce the obtained by the author’s results. For example: PRISMA scheme using Scopus, WoS and PubMed or, Methodi Ordinatio by Pagani et al. (2015, 2017), etc. Without it, it is not known how the authors selected and summarized existing works related to the object of research, the results of this work cannot be reproduced and paper in general is very subjective.
3. You should use 3-5 years old papers for provide modern and usefull analysis in the branch of the investigations.
I cant accept this paper in the current form.
Author Response
Reviewer 2:
Dear authors,
thank you for the improvement of your paper.
- Remainder - structure of the paper. This information should be added at the end of the Introduction section.
This part was added in the previous version of the paper at the end of the introduction section.
- Modern papers with Literature review always used some methodology that help to reproduce the obtained by the author’s results. For example: PRISMA scheme using Scopus, WoS and PubMed or, Methodi Ordinatio by Pagani et al. (2015, 2017), etc. Without it, it is not known how the authors selected and summarized existing works related to the object of research, the results of this work cannot be reproduced and paper in general is very subjective.
I cannot agree with the reviewer, up to date I published several reviews in different journals, and the methodology was not included. It is clear from the discussion contained in the manuscript that the methodology Is based in an analysis of the literature about the text, but introducing a discussion about this aspect in the text does not improve the scientific discussion of the work. Actually, it is not needed any reproducibility of the results of the current manuscript because there are no new results, the work presents a revision of the current status of a research topic.
- You should use 3-5 years old papers for provide modern and usefull analysis in the branch of the investigations.
We do not agree with the reviewer, this review deals with an emerging topic and the reduction of the temporal window to 3-5 years does not provide a true context of the current advances on the discussed topic.
I cant accept this paper in the current form.
We thank to the reviewer for the comments, they were very useful for improving our work.